# Pathogen Safety Issues Around the “Blood Scandals” 1995–2024—A Perspective Built on Experience

**DOI:** 10.3390/pathogens14090868

**Published:** 2025-09-01

**Authors:** Albert Farrugia

**Affiliations:** Medical School, The University of Western Australia, 35 Stirling Highway, Crawley, Perth, WA 6009, Australia; albert.farrugia@uwa.edu.au

**Keywords:** HIV, HCV, pathogen safety, viral inactivation, blood screening

## Abstract

This paper addresses issues around the viral safety of plasma derivatives, which have led to a spate of public inquiries over the past thirty years. These inquiries have ensued following the infection of recipients of plasma derivatives and have focused on identifying which, if any, parties were responsible for these events. The most recent of these inquiries—the Infected Blood Inquiry in the United Kingdom—ran between 2018 and 2022, and has reached conclusions regarding the allocation of responsibility, some of which are discussed in this review. The published reports of the inquiries, supplemented by evidence sourced from the peer-reviewed literature, the policies of government agencies, and public reactions to these processes, form the basis of this review. In addition, the perspective of the author, who has a background in plasma fractionation science as well as being a recipient of plasma products during the period covered by these various inquiries, is offered as a way of augmenting the issues covered. The benefits arising from these, occasionally controversial, inquiries are described, including the heightened commitment to blood safety by policymakers, the embedment of precautionism as a safety principle, and the need for transparency and informed consent in patient management.

## 1. Introduction

The evolution of the safety of plasma derivatives from pathogen transmission has been described previously in this journal [1]. Before the establishment of the current regulatory framework overseeing their safety, pathogen safety issues assumed a secondary role in the supply of these medicines, as issues of sufficiency for treating the severely affected patients with rare chronic plasma protein deficiencies were more dominant in the considerations of industry, treaters, and patients. This was particularly the case for people with haemophilia and allied disorders, who experienced a substantial increase in life expectancy and quality over the 1970s as coagulation factor concentrates manufactured from plasma became increasingly available [2,3]. As a result, an element of complacency, together with a reluctance to introduce measures which were anticipated to affect product supply, became embedded in the therapeutic landscape by the time that the viral epidemics appeared. This was to have tragic consequences.

## 2. The Guilty Pathogens

The blood supply is vulnerable to any microorganism that enters it and is able to maintain its presence in the circulation and in donated blood prior to its being transfused. For the purposes of this paper, the pathogens reviewed are the viral infections forming the focus of attention of the various inquiries. These include the hepatitis viruses B and C (HBV and HCV) and the human immunodeficiency virus (HIV). Other viruses have been transmitted by blood and blood products, including hepatitis A [4], hepatitis E [5], and human parvovirus B19 [6], but have not featured significantly in these processes. This is probably because transmissions by these agents, which have become extremely rare as viral inactivation processes have been established, have not impacted the patient population to anything like the extent of the HIV, HCV, and HBV epidemics.

In his history of viral hepatitis, Schmid [7] shows how early and frequent examples of hepatitis outbreaks were misinterpreted so that an aetiology of a parenterally transmitted virus continued to elude pathologists for over sixty years. During World War II (WWII), examples of deliberate transmission of infectious hepatitis permitted the distinction between an acute, usually non-parentally transmitted hepatitis A and a slower, parenterally/sexually transmitted serum hepatitis B [8], a classification augmented by the demonstration of the immunological distinction between the two types of hepatitis [9]. The emergence of serum hepatitis B as a major hazard of blood transfusion among battlefield casualties in WWII was followed by decades of meticulous investigation before the virus was identified, and its fortuitous property of shedding large amounts of an associated “surface antigen” led to the first, and still the major, screening test for excluding infective blood donations [10]. Together with the identification of donor behaviours associated with infectivity, the introduction of successive generations of screening tests led to the virtual elimination of HBV transmission through blood transfusion by 1995 [10].

Concurrent recognition that not all cases of transfusion-transmitted hepatitis were due to HBV led to the search for another agent(s), designated as non-A, non-B hepatitis (NANBH) [11]. In a triumph for molecular virology, most of these cases were ascribable to a virus characterised in 1989 as a newly identified hepatitis C virus, similarly transmitted parenterally/sexually [12]. The discovery and molecular characterisation led to the rapid development of successively more sensitive screening tests, resulting in the virtual elimination of transfusion-transmitted hepatitis in Western economies (Figure 1) [13], although the problem persists in resource-limited countries [14].

The early years of the Acquired Immunodeficiency Syndrome (AIDS) epidemic and the discovery of HIV have been widely reviewed [15]. Similar to HBV and HCV, the identification and exclusion of donors with high-risk behaviours, followed by the development of screening tests, rapidly largely eliminated the risk of HIV transmission from blood transfusion. The role of understanding the epidemiology of the virus cannot be overemphasised, as this enabled the exclusion of high-risk donors, which led to greatly enhanced transfusion safety before the availability of screening tests (Figure 2) [13].

Figure 3 summarises how these measures of donor selection and screening led to the embedment of a high, although not absolute, level of blood safety for transfused blood and its components produced in blood banks [16]. These measures comprise the first two “pillars” of the “blood safety tripod” described previously [1]. How important these measures were for the safety of plasma derivatives is relevant to the issues assessed by the blood safety inquiries.

## 3. The Third Pillar

The pivotal role of pathogen elimination—the third “pillar” of the “tripod” which also includes donor selection measures and screening of donations—has been discussed in detail [1]. The introduction of validated steps in the manufacture, dedicated specifically to the inactivation and/or removal of contaminating pathogens, has resulted in a situation whereby the current generation of plasma derivatives has a higher safety profile than that of red cell and whole blood recipients, where transfused components are not subject to these processes. The situation prior to the introduction of viral inactivation was the converse; the enhancement of blood safety, through selection and screening processes described above, did not have a similar effect on the safety of plasma derivatives. This is because of the pooling of plasma units as a necessary prelude to the process of industrial plasma fractionation. As previously described, even the small number of infectious units remaining after the selection and screening processes was sufficient, in the absence of viral inactivation, to infect patients given multiple infusions during the early period of the viral epidemics [17]. Comparing the prevalence of infection in the chronically treated haemophilia population (Figure 4) [18] to the residual risk for the transfused population shown in Figure 1 and Figure 3 demonstrates that the haemophilia population remained at risk for a longer period, until the range of viral inactivation methods had been developed and implemented to the extent required. Data extracted from these three sources suggests the following relative risks shown in Table 1.

During the period 1982–1985, the first virally inactivated concentrates were made available in the United States. The processes used were intended to inactivate the hepatitis agents, and as can be seen, they were not very effective in this regard. Infection in haemophiliacs continued to exceed that in the transfused population. Fortuitously, the processes were more effective against HIV, which disappeared from the haemophilia population after 1985. It is important to note that the residual amount of HIV in the plasma supply over 1982–1985, although small, was still enough to infect multiply treated haemophiliacs in the absence of viral inactivation. The risk of both hepatitis viruses remained high for the haemophilia population while continuing to drop for the transfused population as a result of the increasingly effective selection/screening techniques. It is noteworthy that as of 1995, the risk of the hepatitis viruses had also become vanishingly small for the haemophilia population, while a residual, albeit small risk was present for the transfused population. As the introduction of robust viral inactivation steps, such as solvent-detergent treatment [19] and nanofiltration [20] were introduced for plasma derivatives, the current era of safety ensued. The delineation of the timeline of the introduction of the safety measures is important when assessing the inquiries, to which I now turn.

## 4. The “Blood Scandal” Inquiries—Some Examples

A number of inquiries have been carried out over the past thirty years.

**Canada**. The Commission of Inquiry on the Blood System in Canada, more commonly referred to as the Krever Commission or Krever Inquiry, was a royal commission of inquiry into the “tainted blood scandal”, established in 1993 to investigate how the Canadian Red Cross and the provincial and federal governments allowed contaminated blood into the healthcare system [21].

**United States**. In 1993, the Department of Health and Human Services requested that the Institute of Medicine (IOM) establish a committee to study the transmission of HIV through the blood supply. The IOM reported its findings in 1995 [22], putting forward several recommendations for the Department’s consideration.

**Ireland.** The Lindsay Tribunal was set up in Ireland in 1999 to investigate the infection of haemophiliacs with HIV and hepatitis C from contaminated blood products supplied by the Blood Transfusion Service Board [23].

**France**. A series of litigations following the delay of introducing an AIDS test and heated FVIII in order to benefit French companies led to the conviction of several decision makers within the blood service and the political system, with the imprisonment of two leaders of the blood system [24].

**Australia**. Following a report on a particular incident involving HCV in plasma fractionation [25], whose sole recommendation was around the governance of the blood system, the Australian Senate commissioned a committee of inquiry on “*Hepatitis C and the blood supply in Australia*” [26], delivering a report in 2004 which made a few recommendations.

**Scotland**. The Penrose Inquiry was the public inquiry into hepatitis C and HIV infections from NHS Scotland treatment with blood and blood products was established in 2008 and reported in 2015, in a process which was not endorsed by many of the stakeholders [27].

**England**. The Infected Blood Inquiry (IBI), set up in England and lasting from 2018 to 2024, is the most recent of these processes [28]. A number of papers have been published in the scientific literature regarding this process [29,30,31,32]. This process evoked controversy and was characterised by an adversarial attitude and a search for blame regarding the transmissions by viruses in the period described above.

Other processes occurred in a number of countries [33]. This paper will focus on some issues common to the inquiries conducted by public processes as listed above. It is noteworthy that other countries, outside of the Western geographies, experienced similar blood infectivity catastrophes. These have included Romania [34] and China [35]. The scale of these events exceeded, if anything, the events covered by the Western inquiries.

## 5. Some Common Features in “Blood Scandal” Inquiries

This paper will focus on the following issues common to most of the inquiries that have relevance to the issue of pathogen safety. These include the following:

The timeline of the plasma industry’s progression to products safe from viral transmission, both in the commercial and the publicly funded agencies;The issue of self-sufficiency in plasma products;The exposure of patients to clinical studies;The development of regulatory oversight of the industry;Informed consent and patient reactions.

These will be assessed sequentially.

### 5.1. The Progression Towards Viral Inactivation

This issue has been described [1]. The first plasma derivative subject to pathogen activation was albumin solution, which was heated for 72 h at 60 °C in order to eliminate hepatitis transmission. Hepatitis was recognised as a consequence of haemophiia replacement therapy from the outset of this treatment [36], but treater opinion was that the benefits of treatment outweighed the risk [37]. Most cases of confirmed hepatitis B resolved without sequelae, and early studies on Non-A, Non-B Hepatitis (NANBH—subsequently shown to be mostly due to hepatitis C Virus—HCV) indicated a benign, non-progressive course [38]. This optimistic picture was disrupted by several studies indicating a progression to cirrhosis in a high proportion of patients. Mannucci’s review is recommended for a full recount of this area [37]. This growing realisation of the hepatitis problem initiated the first efforts to include viral inactivation in the manufacture of factor concentrates. The first such endeavour in the late 1970s subjected FVIII in solution to heat treatment at 60 °C for 10 h, a process known as pasteurisation and an intuitive measure considering the known efficacy of this step in making albumin safe [39]. Despite the addition of stabilisers to protect the FVIII from denaturation, the overall yield of this process was less than 15% over the starting plasma, and the amount of product manufactured could be used for treating only a very small population of previously untreated patients (PUPS). Studies in these patients indicated that the product did not transmit any of the transfusion-transmitted viruses [40]. Efforts in the early 1980s by the commercial industry in the USA, initiated because of the hepatitis risk but influenced by the growing awareness of the AIDS epidemic in haemophiliacs, led to the licensure of a number of heat-treated FVIII concentrates, which were heated under such conditions that the FVIII activity could be preserved [22] (Table 2).

Although the timeline of these processes indicates clearly that they were directed against hepatitis and not AIDS, the viral nature of which was only confirmed in 1984 [41], and some of these concentrates were shown not to transmit HIV or AIDS to haemophilic patients [42], although cases still occurred with others [43]. In some instances, companies retained unscreened plasma for the production of FVIII concentrate after the introduction of screening tests for HIV, resulting in viral loads in the plasma which could not be eradicated by the early forms of heat treatment [44,45]. By 1988, 18 patients treated with this form of heat-treated concentrate had become infected with HIV [22], and the manufacturer withdrew the product from the market and modified the procedure to 68 °C for 72 h. With the exception of the pasteurised FVIII concentrate alluded to above, all these early forms of heat-treated products continued to transmit HCV. The introduction of solvent-detergent treatment of coagulation factors, first described in 1985 [46] and clinically validated in 1988 [47], finally eradicated the scourge of hepatitis transmission in haemophilia.

Consideration of these timelines is important when assessing the reaction of the various inquiries to the industry’s timeliness, or otherwise, in introducing pathogen safety measures. Some inquiries addressing the role of publicly owned fractionation companies have been extremely critical in this regard, particularly the IBI in the UK. The relevant timelines for the introduction of viral inactivation in Scotland [48] and England [49] provide detailed records of the progression towards viral inactivation for the haemophilia treatment products in these countries. Similarly, the Australian process [25] provides information for the then publicly owned fractionator in that country. Table 3 extracts this information.

It is clear from this timeline that the British public fractionators did not lag significantly behind the commercial sector when introducing effective measures to inactivate HIV in FVIII. Similar endeavours to inactivate HCV were, at least, concurrent with the commercial industry’s rapid adoption of solvent-detergent treatment. The frequently cited pasteurisation process, during this period, was attaining FVIII yields which, if adopted by the public system in the UK, would have denied treatment to the vast majority of haemophilia A patients. It is noteworthy that such a process was not rejected out of hand, but was adopted for preliminary clinical investigation [53] (Chapter VII in [54]) before being overtaken by other approaches more conformant with ensuring adequate supply as well as safety. Efforts to inactivate HCV, hampered by the lack of certainty regarding the properties of this agent, were not significantly behind those for HIV, and, again, concurrent with the commercial industry’s achievements. The public fractionators were, justifiably, concerned regarding the effect of viral inactivation on FVIII manufacturing yields, a lesser issue for the commercial sector’s ability to offset any such losses by recruiting more paid plasma donors. Several of the inquiries have criticised the public authorities for not striving for higher levels of self-sufficiency in plasma products as outlined below. It is pertinent that the adoption of the very first reported viral inactivation processes, characterised by very low yields, would have made self-sufficiency much harder to attain.

It is evident that efforts to inactivate viruses in FIX concentrates, which were also enriched in the related factors II and X, lagged behind those for FVIII, hampered to some extent by perceived evidence of enhanced thrombogenicity when these products were heated [53]. Unfortunately, this was based on the mistaken hypothesis that activated coagulation factors, whether generated through heating or not, were responsible for FIX concentrate thrombogenicity. It was only when this concept was disproved through the development of a single FIX concentrate purified from the other zymogens that it was shown that these were the main contributors to thrombogenicity [55]. It is possible that the meticulous line of investigation described and followed by the British fractionators in order to minimise thrombin generation during heating [32,49,56] resulted in an unnecessary delay in the introduction of severe heat treatment for FIX concentrates.

### 5.2. The Issue of Self-Sufficiency in Plasma Products

In several of these inquiries, the public health authorities, including the fractionation companies then owned by the state, were admonished for not making sufficient efforts to approach a situation of national self-sufficiency in plasma products. The claim has been made repeatedly that such a policy, restricting use to products acquired from a national source of plasma with a safer epidemiological profile than that of the paid donor plasma of the commercial sector, would have minimised infection in the recipient community.

This work disputes this assertion. Considering the epidemiology of the viruses concerned and the necessity, imposed through Good Manufacturing Practices, of pooling plasma of thousands of donors to ensure batch homogeneity and sampling, the restriction of plasma source to domestic donors would have had little or no effect. Using the principles described by Lynch et al. [17], it can be estimated that, in the absence of viral inactivation, infection would have occurred, even at infectious agent prevalences that were lower than those from the US paid donor population. This is best exemplified by the case of Australia, a country which restricted concentrated provision to domestic products manufactured during the period of interest by a state-owned facility. This restriction did not protect a large proportion of the spectrum of patients with bleeding disorders from becoming infected with HIV [57]. It is noteworthy that 25% of patients with von Willebrand’s Disease treated solely with blood bank cryoprecipitate, an inadequate modality but one touted by the English IBI as being of superior safety, also developed HIV. Assessing the issues raised during the process in France, the consequences of the delay in screening plasma for HIV and the retention of unscreened plasma used in the manufacture of FVIII by the domestic system make for sobering reading [24,58]. Australia’s policy of restricting entry for treatments sourced from overseas appears to have had little effect in speeding up progression to virally inactivated products (Table 3).

### 5.3. The Exposure of Patients to Clinical Studies

A particular area of criticism, in particular in the English IBI, has focused on the clinical trials carried out to optimise haemophilia treatment during the pre-viral inactivation era. These included the trials to assess the effects of prophylactic therapy in haemophilia, drawing on large and regular doses of FVIII in order to compare such preventive treatment to episodic therapy, which uses much lower amounts of product [59]. These early, pioneering efforts were, like all therapeutic interventions in this period, tragically associated with a higher risk of viral infection, and the responsible clinicians were heavily criticised by the IBI for using patients as “research subjects” and by the popular press as “guinea pigs” [60]. These comments appear to undermine the basic principle of evidence-based medicine, in that the clinical trials were carried out according to randomised clinical trials, and preceded by forty years of subsequent trials, which established the benefits of prophylaxis [61]. The use of patients to assess the viral safety of concentrates in previously untreated patients (PUPs) was likewise criticised, with allegations that patients were used instead of chimpanzees for this purpose [62]. The use of chimpanzees to assess product safety from hepatitis transmission has been shown to be fallible in many instances [63], and the peer International Society for Thrombosis and Haemostasis had specified that evidence for safety from hepatitis transmission had to be obtained from unexposed patients [40,64], a principle accepted by the regulatory authorities. These facts were ignored by the IBI.

### 5.4. The Development of Regulatory Oversight of the Industry

Several of the inquiries—Canada, Ireland, USA—voiced criticism of their perceived lack of adequacy by government regulators in overseeing the safety of plasma and plasma-derived medicines. The inquiries maintained that, in many instances, intervention was late and inadequate. Issues of resource inadequacy were described to explain part of this perception. In addition, the difficulties in overseeing the blood system in many countries, particularly federations such as Canada and Australia, where much of this activity was devolved from the central government to the individual jurisdictions constituting the federation, were described. In these countries, the central government had few if any powers to intervene in the blood service’s activities, e.g., the selection of low-risk donors. The benefits of the inquiries included legislative changes to address this problem, and recommendations to increase resources in the regulatory authorities, recommendations which, in most instances, were at least partially met.

Two fundamental features ensued because of these criticisms. The first addressed the perceived procrastination by regulators in the face of emerging, still poorly understood, pathogenic infections. As pointed out by the American process (page 14 in [22]):

“Where uncertainties or countervailing public health concerns preclude completely eliminating potential risks, the FDA should encourage, and where necessary require, the blood industry to implement partial solutions that have little risk of causing harm.”

This point addresses the propensity of some regulators and parts of the industry to delay action until full certainty is obtained regarding the scope of the infectious risk problem. For example, the delay in assessing causality, using conventional criteria such a Koch’s postulates [65], until the virus was fully characterised, to HIV as the cause of AIDS was used to excuse some of this delay. The same issue pertains to HCV.

This criticism has led to the second regulatory fundamental consequence of the blood infection inquiries, in the embedding of the so-called precautionary principle in blood regulatory policy. This principle specifies the approach to be taken in the face of emerging, as yet uncertain risks, to the safety of the blood supply, and states that:

“When an activity raises threats of harm to human health or the environment, precautionary measures should be taken even if some cause and effect relationships are not fully established scientifically. In this context the proponent of an activity, rather than the public, should bear the burden of proof.”[66]

It is of interest that in his report on the Canadian blood inquiry, Justice Horace Krever did not mention the precautionary principle once; however, his words on how authorities should react to emerging blood safety threats bear noting:

“Preventive action should be taken when there is evidence that a potentially disease-causing agent is or may be blood borne, even when there is no evidence that recipients have been affected. If harm can occur, it should be assumed that it will occur. If there are no measures that will entirely prevent the harm, measures that may only partially prevent transmission should be taken.”[67]

Whether intentionally or not, these words summarise the complex infrastructure of regulatory and other public health measures which underpin the blood system today and contribute to its safety. Since this precautionary approach has resulted in the successive introduction of measures, many of them expensive, intended to prevent and eliminate pathogens from the blood supply, it has attracted critics, particularly from economic rationalists [68], and continues to be debated with vigour. This author was responsible for implementing this principle during his engagement in the Australian Therapeutic Goods Administration (1994–2008) and hopes that its continued embedment in blood regulation is not displaced, if the errors of the past are not to be repeated. The role of successively developed frameworks, including risk-based decision making [69], is acknowledged, and their uptake by the current generation of regulatory authorities is commendable; however, as is pointed out below, the non-linear and unpredictable nature of blood safety threats [70] suggests that precautionism should be retained as a policy principle.

### 5.5. Informed Consent and Patient Reactions

Current established principles of informed patient consent [71] were less emphasised in the era of nascent haemophilia care, which constituted the focus of interest of the blood safety inquiries. The intimate relationship between haemophilia carers and their patients, often mediated through the patients’ parents, may have contributed to this. Furthermore, the nature of acute bleeding episodes left less room for an informed conveying of the possible adverse events of treatment, especially when considering that the full extent of these was incompletely and imperfectly known by the treaters. This issue was visited by several of the inquiries. Patients who were infected by pathogens as a result of the treatments, or their relatives, asserted vehemently that consent was not obtained for their possible exposure to products associated with HIV, HCV, and other viruses. This criticism was directed against all those involved in delivering care, including nurses [72]. The warnings regarding pathogen transmission incorporated in the packaging materials of the products from the mid-1970s onwards were judged to be inadequate measures for such consent.

It is evident and regrettable that this principle was largely ignored during the era of haemophilia care, scrutinised by the various inquiries. What is less evident is the extent to which a higher level of such consent may have made a difference to the patients involved, in terms of morbidity and mortality. Those heading the IBI, for example, allege that the absence of a process of informed consent led to two deleterious outcomes:It disallowed patients, or in many instances, their parents, from opting out of the treatment offered, because of the possibility of pathogen transmission;It obviated the possibility of alternatives to the treatments offered, again leading the patients to an option which led to infection.

This author has nothing but respect for the individuals who suffered and, in many cases, died from the infections described in this paper. However, he feels a right, as a patient himself who acquired two blood-borne infections from factor concentrates [73], to express a view based on his personal experience, which is necessarily subjective. My parents, faced with seeing their child in the agony of joint bleeds in the era of no treatment, would not have hesitated to alleviate my suffering irrespective of what were then ill-perceived risks of possible infection. At a more mature age, the life-threatening episode which led to my first exposure and subsequent infection with hepatitis left me with little choice but to acquiesce to the treatment which saved my life. In the era preceding the development of safer plasma concentrates, to be rapidly succeeded by the current and vibrant era of therapies not derived from plasma, I propose that there was little real choice.

## 6. Final Reflections

The tragedy of the viral epidemics in the population of recipients of plasma products during the 1970s to the 1990s is all the more poignant because it devastated a community of patients with rare diseases, which had, before the development of these products, made their lives short and full of pain. The inquiries have served several purposes. They have generated an impressive volume of documentation, hitherto unavailable, which is now in the public domain and available for further research and reflection. They have provided a synthesis of this information, which, for those interested in understanding the history of this therapeutic era, is an invaluable resource. And they have resulted, in some instances after long and painful periods of government indifference and procrastination, in securing compensation for some of the victims of the epidemics, so that they may continue their lives with some level of dignity and comfort.

In addition, the public interest in the safety of the blood system has increased greatly as a result of these inquiries, and, in its turn, this has influenced public policymakers in their oversight of the blood system. This influence has contributed to the continued implementation of blood safety measures through the precautionary approach alluded to above. Although the inquiries have paid scant attention to the other pathogens threatening blood safety which have emerged, and continue to emerge, since the era covered by their remit, the heightened awareness generated by the blood viral epidemics in the transfused population has ensured that a robust system for minimising pathogen transmission through blood has been established in the high-income countries of the Western world. Although less prevalent in public discourse than HIV, HCV, and HBV, the additional infectious agents which have threatened the blood supply have been largely excluded as a threat to plasma product recipients, mostly thanks to viral inactivation processes. This situation is regrettably not reflected in the situation in low and middle-income countries, where the safety of mainstream transfusion is still threatened, and where access to plasma protein therapies is minimal because of economic issues.

But these achievements, fundamental as they are, do not constitute the most important outcome of these inquiries. This should be the warning that, in the absence of constant vigilance and scientific rigour, these catastrophic events can happen again. The current era of haemophilia care, in particular, is characterised by the availability of a virtual smorgasbord of treatments, sharply contrasting with the paucity of alternatives available in the era preceding and during the viral transmissions. These treatments impose increasingly complex interventions such as gene therapy and manipulations of the haemostatic system [74]. Robust pharmacovigilance systems monitor their effects on patients [75]. But the non-linear nature of health care interventions [76] continues to impose uncertainty, as does the inherent complexity of biological systems, the limitations of medical knowledge and evidence, and the variability among individual patients and their unique circumstances [77]. This imposes a need for precautionism in blood safety and other areas. If we have not learnt this, the inquiries described in this article would have been undertaken in vain.

## Figures and Tables

**Figure 1 pathogens-14-00868-f001:**
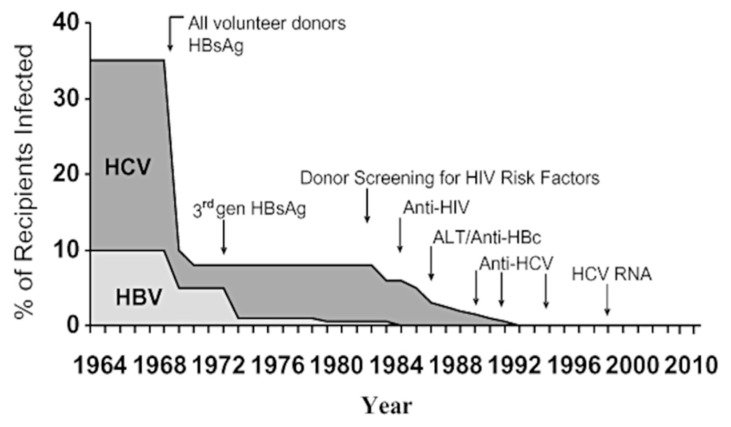
Declining incidence of transfusion-associated hepatitis in transfusion recipients monitored prospectively at the NIH Clinical Center. Incidence of hepatitis, traced from 1969 to 1998, demonstrated a decrease in risk from 33% to nearly zero. Arrows indicate the main interventions in donor selection and screening that affected this change. Reproduced from Reprinted with permission from Ref. [13]. 2010, John Wiley and Sons.

**Figure 2 pathogens-14-00868-f002:**
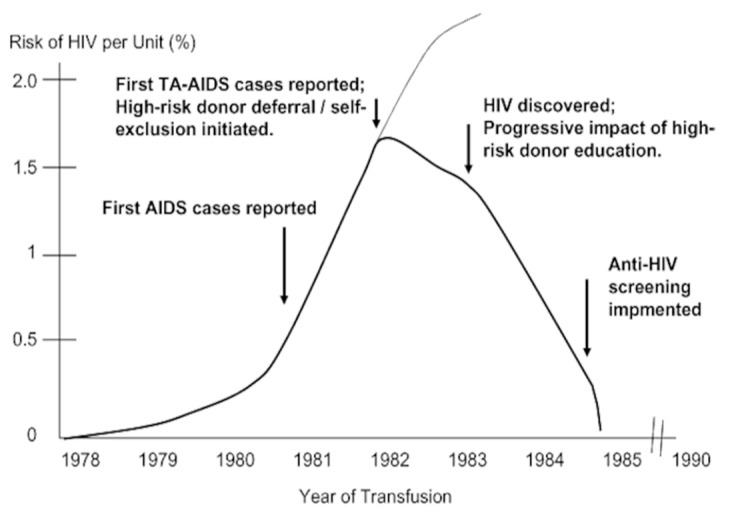
Risk of HIV transmission by blood transfusion, before the implementation of HIV-1 antibody screening. The results of the risk projection demonstrate the dramatic decline in HIV risk coinciding with progressive implementation of high-risk donor qualification and deferral measures and preceding the availability of prospective antibody screening. TA = transfusion-associated. Reprinted with permission from Ref. [13]. 2010, John Wiley and Sons.

**Figure 3 pathogens-14-00868-f003:**
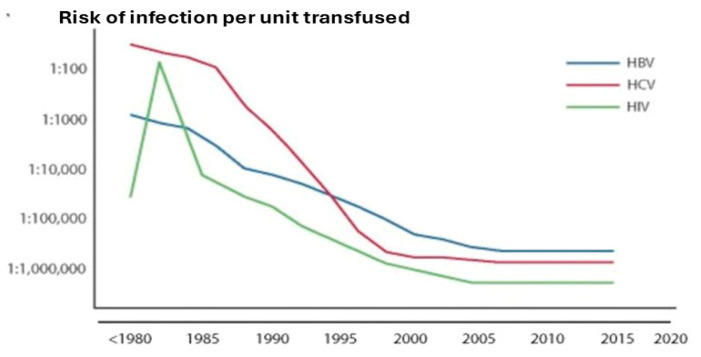
Risks of major viruses, per unit infectious risk for HBV, HCV, and HIV from 1980 to 2018. Reprinted with permission from Ref. [16]. 2019, Elsevier.

**Figure 4 pathogens-14-00868-f004:**
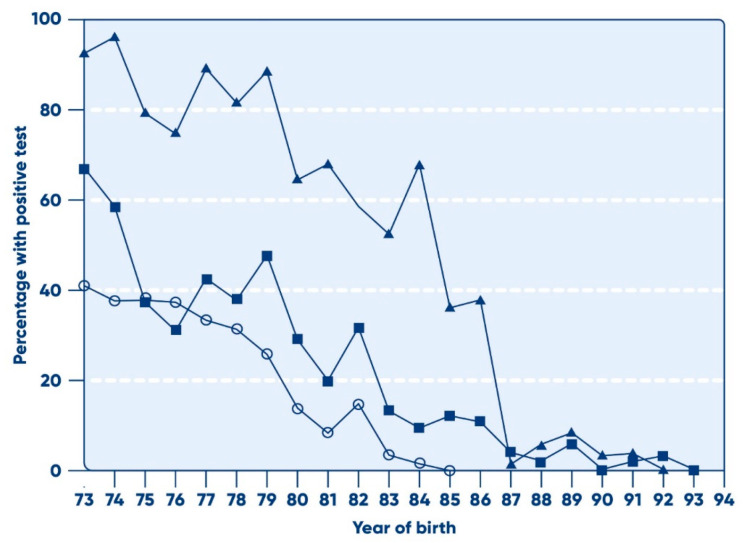
Prevalence of blood-borne infections in haemophilic birth cohorts in the United States. Based on the results of laboratory testing for HBV (◼), HCV (▲), and HIV-1 (○). The proportion was zero for HIV after 1984, for HCV after 1992, and for HBV after 1993. Reprinted with permission from Ref. [18], 2001, John Wiley and sons.

**Table 1 pathogens-14-00868-t001:** Infection risks in the transfused and haemophilia population in the USA.

% Recipients Infected	Transfused Population		Haemophilia Population		
1982	Hepatitis (All Types)	AIDS/HIV	HBV	HCV	HIV
1982	10	1.5	30	60	18
1985	8	0.25	15	38	0

**Table 2 pathogens-14-00868-t002:** Early attempts at heat treatment by the commercial sector—from [21].

Plasma Fractionator and Method for Heat-Treated Factor VIII Concentrate	Date Applied for FDA Licensing	Date License Granted by FDA
Baxter Healthcare (dry heat, 60 °C for 72–74 h)	June 1982	March 1983
Miles, Inc. (formerly Cutter Biological)		
(i) (liquid pasteurisation, 60 °C for 10 h) (ii) (dry heat, 68 °C for 72 h)	August 1983 November 1983	January 1984 February 1984
Alpha Therapeutics (wet heat, 60 °C for 20 h)	December 1982	February 1984
Armour Pharmaceutical (dry heat, 60 °C for 30 h)	December 1982	January 1984

**Table 3 pathogens-14-00868-t003:** The author’s assessment of the progression towards viral inactivation in some publicly owned fractionation companies. Extracted from [25,48,49].

Country	Year in Which Virally Inactivated FVIII Was Issued		Year in Which Virally Inactivated FIX Was Issued	
	Process Which Could Destroy HIV [50]	Process Which Could Destroy HIV, HBV, and HCV [51]	Process Which Could Destroy HIV, HBV and HCV	Source
Scotland	Q4 1984	Q4 1986	Q3 1985	Ref. [48]
England	Q3 1983 (only 3 patients treated)	Q3 1985	Q4 1985	Ref. [49]
Australia	Q4 1984 [50]	ca Q1 1989	Q4 1984 (HIV only) Q2 1993 (special access) Q2 1998 (market approval)	Ref. [52] Personal information from author

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
