# Peer review of "Pathogen Safety Issues Around the “Blood Scandals” 1995–2024—A Perspective Built on Experience"

_pathogens, 2025, doi:10.3390/pathogens14090868_

Round 1

Reviewer 1 Report

Comments and Suggestions for Authors

Summary

This is a timely and well-written narrative review that brings together technical history, policy analysis, and a personal perspective to address pathogen safety of plasma-derived products and the inquiries that followed major viral epidemics from the 1970s onwards. I found the dual perspective (scientist and patient) to be one of the strongest features of the paper, giving it both authority and originality. The manuscript is very much in line with the scope of Pathogens and will likely be of interest to clinicians, transfusion specialists, and policy makers.

Overall recommendation

The science and conclusions are sound; the main improvements would be around clarifying sources in the timelines, softening a few subjective passages, and making sure figures/permissions are fully in order.

Major comments

  1. Table 3 / timelines
    • The timelines are very useful, but at times it is difficult to see where dates come from. I would suggest adding a clear source for each date (in a separate column or as footnotes).
    • Date format should be consistent (ideally YYYY or Q# YYYY), with “c.” for approximations. Where different sources disagree, just explain briefly the rule you adopted (e.g. “earliest verifiable date”).
    • A brief one-sentence description of how the dates were collected would help transparency. A compact visual timeline figure (1975–2024) would also be very effective alongside the table.
  2. Tone of first-person sections (pp.9–11)
    • The author’s personal reflections are a real strength, but occasionally the language feels quite absolute. It would read more balanced if those statements were explicitly framed as opinion/experience, or linked where possible to supporting evidence (patient surveys, inquiry reports). This way the personal voice is preserved but perceived bias is reduced.
  3. Figures and permissions
    • Please make sure that for any reproduced figure (e.g. Busch et al.), the permissions or licence for reuse are clear. If they cannot be obtained, a re-drawn schematic with proper attribution is a good solution.
  4. Methods / sources
    • It would help to include a short sentence early on explaining the scope of the source search (public inquiries, manufacturer submissions, peer-reviewed literature, etc.). This makes the review process more transparent.
  5. Perspective beyond viral agents
    • A short paragraph could acknowledge the situation in low- and middle-income countries, and briefly mention why agents such as vCJD, HEV/HAV, or parvovirus B19 are not a main focus here (or, if relevant, say a word on them). Even a few lines would broaden the appeal without lengthening the paper much.

Minor comments

  1. There are a few spelling slips (e.g. haemophiia → haemophilia, pateurised → pasteurised, sequalae → sequelae, “in a AIDS” → delete “a”).
  2. Please define abbreviations consistently at first use (e.g. NANBH → define once, then use HCV).
  3. The abstract could be expanded by one or two sentences to state the main take-home message (timeline findings, precautionary principle, lessons learned). Adding keywords such as “blood-safety inquiries”, “precautionary principle”, and “plasma fractionation” would improve discoverability.
  4. Double-check all recent references (2024–2025), including DOIs and publication status.
  5. Table 1: please make denominators/units explicit for infection rates.
  6. Figure captions: list the source and permission/licensing status.
  7. On p.8, the section on self-sufficiency might benefit from a short illustrative example of pooling mathematics.
  8. On p.9–10 (informed consent), tone down some of the absolute statements and, where possible, add references (surveys, inquiry data).
  9. When saying transfusion-transmitted hepatitis was “virtually eliminated” in Western countries, please indicate a clear timepoint (e.g. early 1990s) and cite accordingly.

Concluding note

Overall, I think this is a very strong and valuable manuscript. With a few small adjustments for transparency and balance, it will make an excellent contribution. 

Reviewer 2 Report

Comments and Suggestions for Authors

This manuscript is by one of the most active and acknowledged scientific debaters in the field of infectious safety of blood based pharmaceuticals. It is at some points very personal - with reason - so it is not straightforward to evaluate it as a scientific paper. I have tried to focus my observations on the general principles of scientific writing and the context of blood supply, not expressing my opinions.

The manuscript (Manuscript or MS) starts with Introduction where the perspective of plasma based pharmaceuticals is imposed. SInce the Title of the article includes the "Blood Scandals" I would expect to see the Scope of this MS described in the Introduction. For me it is completely acceptable to restrict this MS to the plasma based pharmaceuticals and testing but the "infected blood" inquiries and the backgrounds to the unfortunate events include many aspects of blood donor selection and philosophy of blood supply. These need not be taken into this review but the Author should open the review with a definition of scope.

Another general comment concerns the chapter on Regulatory Oversight. As I understand, the Author makes a case for a strong commitment to Precautionary principle (rows 330-332). It would be justified to mention that in the discussion on this matter, alternatives like the Risk Based Decision Making (RBDM, Risk‐based decision making in transfusion medicine - Leach Bennett - 2018 - Vox Sanguinis - Wiley Online Library) have been introduced.

Detailed comments by row (r):

(r36-38) There is something in this sentence which I don´t understand. Should it read Other viruses which have been...? "Perhaps they should have." is also a bit vague and there is no rationale for this. Historically they did not.

(Fig 1, caption) It seems to me that this is from Perkins and Busch 2010. Busch et al was from 2019.

(Fig 2, caption) This seems to be from Perkins&Busch 2010 where it was borrowed from Busch et al 1991.

(r 82) There is now a pathogen inactivation process for platelet components. It is may be true that still the plasma derivatives have a higher safety profile but the wording might need adjustment.

(r 93) I don´t understand how the Figs 1 and 3 demontrate the difference between persons with haemophilia and other recipients of blood products.

(r 93) Which sources? The articles where the Figs 1 and 3 came from?

(r 102) New infections disappeared but not the disease?

(Fig 4, caption) I do not find Soucie et al 2001 in the reference list.

(r 151) pathogen safety of plasma derivatives. This is not a list for blood component safety issues.

(r191) incongruent format of citation.

(r28) the reference for Australia (refnr 24) is from 1993 and the history on Australia continues until Q2_1998. How is this possible?

(r209) using the word "significantly" can be challenged since there seems to be 1-2 years gap between the years in Tables 2 and 3. In the context of a blood borne epidemic it seems not too short? There should be a justification for this.

(r223) Using the word "underplayed" is also a value statement with no rationale expressed.

(r235-237) The sentence "Such retrospective regret is of..." might be formulated more neutrally and backed by observations on the changes introduced - or not introduced?

(r260) discarding and using the same plasma is hard for me to understand?

(r 270) "line" should be "like".

(r282-283) Using the word "regrettable" should be backed by more concrete negative consequences.

(r359-372) Here it is clear that the Author describes his own experiences and views. I find this acceptable - and valuable - and see no need to edit.

(r393-394) There are also other properties of medical interventions than non-linearity which continue to impose uncertainty. This wording should be made more general. Another point: I do agree that a high degree of caution, carefulness and oversight is necessary with new and emerging technologies but going up to "precautionism" which has a slightly exaggerated connotation is a matter of opinion. However, I respect the freedom of the Author to express his view and I do not see a need to change this word.

Reviewer 3 Report

Comments and Suggestions for Authors

Dear Sir,
I read with interest and attention the contribution: Pathogen safety issues around the “blood scandals” 1995 – 2024 2
– A perspective built on experience.
By Farruggia A.
Manuscript ID: pathogens-3844911

The author addresses the safety issue of plasma-derived drugs in light of HBV, HCV, and HIV infections not only from an epidemiological perspective but also by considering several of the various investigations conducted in the decade from 1993 to 2022 in several Western countries: Canada, the United States, Ireland, France, Australia, Scotland, and England.

Personally, I found the work extremely interesting.
Of particular interest to me was the discussion rejecting the hypothesis that limiting the source of plasma from voluntary donors could have limited the transmission of HBV, HCV, and HIV infections. This is a strong and absolutely acceptable statement, although it is not politically correct and is in contrast with common thinking.

In my opinion, from a conceptual standpoint, two aspects should be emphasized for fairness.
First, the plasma-derived drugs used in the treatment of hemophilia A (FVIII concentrates) and hemophilia B (FIX concentrates) have radically changed the duration and quality of life of patients.
Second, these investigations have involved almost exclusively Western countries. Nothing similar has been conducted in Eastern Europe (e.g., Romania) or Asia (e.g., China).

On page 5, the author introduces the concept of the third pillar. He should clarify what the first and second pillars are.
